# Patch test results in paediatric patients with atopic dermatitis in Laos

**Catriona I. Wootton**[1,2]*, **Mong K. Sodaly**[3☉], **Somxay X. Billamay**[3☉], **John S. C. English**[4‡], **Mayxay Mayfong**[1,5,6‡]

1 Lao-Oxford-Mahosot Hospital-Wellcome Trust Research Unit (LOMWRU), Vientiane, Laos, 2 Churchill Hospital, Oxford University Hospitals NHS Foundation Trust, Oxford, United Kingdom, 3 Allergy Clinic, Lao-Korea Childrens' Hospital, Vientiane, Laos, 4 Nottingham NHS Treatment Centre, Nottingham, United Kingdom, 5 Institute of Research and Education Development (IRED), University of Health Sciences, Ministry of Health, Vientiane, Laos, 6 Nuffield Department of Medicine, Centre for Tropical Medicine and Global Health, Churchill Hospital, Oxford, United Kingdom

☉ These authors contributed equally to this work.
‡ These authors also contributed equally to this work.
* ciwootton@aol.com

## Abstract

### Background

Dermatological services in Laos, South East Asia are limited mainly to the capital and patch testing is currently not available, so no data exists regarding the common cutaneous allergens in this population.

### Objectives

The aim of this study was to document common allergens in paediatric patients with atopic dermatitis attending the allergy clinic in the capital, Vientiane.

### Patients/Materials/Methods

Fifty paediatric patients with atopic dermatitis were patch tested using TRUE Test® panels 1 to 3 (35 allergens). Readings were taken at Days 2 and 4.

### Results

Twenty-six positive patch tests were recorded on Day 4 in 15 children (30%). The most common allergens were: gold (18%), nickel (10%), formaldehyde (6%) and p-Phenylenediamine (6%). Other positive allergens were potassium dichromate (2%), cobalt dichloride (2%), Bronopol (2%), paraben mix (2%), fragrance mix 1 (2%) and neomycin (2%). The majority of the patients with positive reactions were female.

### Conclusions

This study represents the first documented patch test results in the Lao population. It is hoped that these findings will help clinicians to advise the families of children with atopic dermatitis on common allergens to avoid and inform future work on contact dermatitis in this population.

**Data Availability Statement:** All relevant data are within the manuscript and its Supporting Information files.

**Funding:** CW The British Association of Dermatologists Geoffrey Dowling Fellowship The

funders had no role in the study design, data collection and analysis, decision to publish, or preparation of the manuscript.

**Competing interests:** The authors have declared that no competing interests exist.

## Introduction

Laos is a landlocked country in South East Asia with a population of almost 6.8 million people; roughly 800,000 of which live in the capital, Vientiane. The country is made up of several different ethnic groups and the main occupation is rice farming. A dermatology clinic exists in the capital but patch testing is currently not available. The aim of this study is to document common allergens within the paediatric atopic dermatitis (AD) population. Controversy still remains regarding whether patients with AD are more or less likely to develop contact allergy [1] but paediatric patients with AD were selected for this study as these individuals are exposed to potential allergens from an early age.

Currently there is no data on common contact allergens in the Lao population, so this study along with its sister study looking at contact allergy in an asymptomatic adult population (medical students), aims to establish the common allergens in this community, paving the way for future research.

## Methods

Paediatric patients with AD, known to the allergy clinic at the Lao-Korea Childrens' Hospital in Vientiane, were invited to attend for patch testing. In addition, any new patients presenting with AD were also given the opportunity to take part, S1 Fig outlines the patient selection process.

This study ran from August 2017 to January 2018. Ethical approval was granted by the Lao National Ethics Committee for Health Research. The process of patch testing was explained and images of patch tests being performed were also shown to the patient and their family and verbal consent given by the parent or guardian of the child. Verbal consent was deemed appropriate by the National Ethics Committee for the study as this is standard practice for patch testing in the author's experience and consent was recorded on the study proforma at each visit.

TRUE Test® (SmartPractice, Denmark: http://smartpractice.dk), 3 panels equalling 35 allergens in total were applied to the child's upper back and an additional film dressing was applied if required. S1 Table lists all of the allergens tested. TRUE Test® panels were used due to the need for consistency and precision of the amount of allergen present in the test as well as the environmental challenges especially heat, humidity and transportation issues and lack of facilities for preparing patch tests on site. TRUE Test® panels are not routinely used in the paediatric population but Jacob et al. [2] found these patch tests to be safe and efficacious in the paediatric population.

The participants were reviewed on Day 2 (48 hours) (when the patches were removed) and Day 4 (96 hours). The patch testing was performed and the results interpreted by an experienced dermatologist, following the British Association of Dermatologists guidelines on the management of contact allergy [3]. The participants were given $15 on Day 4 as a contribution towards travel costs.

All data were recorded anonymously using the Open Data Kit programme on tablet devices. The London School of Hygiene and Tropical Medicine server was used for data storage (http://opendatakit.lshtm.ac.uk).

## Results

Fifty paediatric patients were patch tested, ranging in age from 13 months to 14 years, with a mean age of 4.5 years. The majority (62%) of the participants were female. All of the participants had AD, only one also had a formal diagnosis of asthma. The severity of the participants' AD was not formally assessed or recorded, however the majority of the participants would fall into the classification of mild to moderate AD. Fifteen patients (30%) were found to have

positive patch test reactions at Day 4, with 26 positive reactions being recorded in total. The ratio of female:male participants with positive reactions was 11:4. Six patients had two positive reactions, one had three positive reactions and one had four positive reactions. Gold sodium thiosulphate and nickel sulphate were the most common allergens; 9 children (18%) tested positive to gold and 5 children (10%) to nickel. The incidence of gold allergy was more common in girls with a 2:1 female:male ratio, whereas nickel allergy had a more equal incidence in males and females (3:2). Formaldehyde and p-Phenylenediamine each caused positive reactions in 3 participants (6%), all of which were female. Potassium dichromate, cobalt dichloride, 2-Bromo-2-nitropropane-1,3-diol (Bronopol), paraben mix, fragrance mix 1 and neomycin each showed a positive reaction in one case (2%). The patient with 3 positive patch tests reacted to: paraben mix, p-Phenylenediamine and gold sodium thiosulphate, and the patient with 4 positive patch tests reacted to: nickel sulphate, potassium dichromate, formaldehyde and 2-Bromo-2-nitropropane-1,3-diol, again both of these patients were female.

## Discussion

This is the first patch test study to be performed in Laos. The paediatric AD population was selected for several reasons: firstly, children with AD are often exposed to potential allergens from an early age [1]: debate still exists regarding whether AD increases the risk of contact allergy or not. Rodrigues & Goulart's [4] review of patch test results in children found studies both revealing and refuting a statistical difference between patients with and without AD. In addition, as patch testing is otherwise not available to this cohort, uncovering any allergic contact dermatitis and subsequent avoidance of allergens could lead to improvement in the patient's AD symptoms. The final, and not insignificant, reason for selecting this cohort was that these individuals had already demonstrated health-seeking behaviour by attending the clinic and were therefore more likely to be willing to engage in the process of patch testing and attend for the follow-up visits. Our 100% attendance at follow-up appointments, was probably due partly to this fact but also to the financial remuneration they received to help towards travel costs.

Reviewing the literature on patch testing in the paediatric population reveals patch test positive rates ranging from to 25 to 95.6% [4–6]. The most common allergens reported are: nickel, cobalt, fragrance mix 1, potassium dichromate, wool alcohols, Balsam of Peru, neomycin, colophony, thiomersal and methylchloroisothiazolinone/methylisothiazolinone (MCI/MI) [5, 7–14]. Interestingly the most common allergen in our study was gold (18%); gold is rarely reported as a common allergen in other patch test studies. There may be several reasons for this: firstly, gold is present in the TRUE Test® panel 3 but does not feature in some other patch test series, such as the European baseline series and it is only in the last two decades that gold has really been accepted as a potentially significant allergen [15]. Finally, it may be that sensitivity to gold is more common in certain populations, such as southeast Asian and Middle Eastern populations [16–18], where higher rates of sensitivity to gold are reported in the literature. Boonchai & Iamtharachai [18] found a higher incidence of sensitivity to gold (30.7%) compared to nickel (27.6%) in their study of adults in Thailand. Shakoor et al. [16] and How et al. [17] also reported reasonably high incidences of sensitivity to gold in their studies on adults in Saudi Arabia (13.5%) and Malaysia (15.2%). In comparison, Fowler et al. [19] reported the 9.5% of their patients with suspected contact allergy in North America had a sensitivity to gold. Exposure to gold, resulting in contact allergy is believed to come from three main sources: dental fixtures, jewelry and medical use such as medications and coronary artery stents. The traditional and cultural use of gold jewelry (especially for young children) in areas such as South East Asia and the Middle East, may explain the higher incidences of gold allergy reported.

Three participants (6%) were found to be allergic to p-Phenylenediamine (PPD). PPD is not often reported as a common allergen in the paediatric population, this may be because it is not always included in a paediatric series due to the risk of sensitisation. One study, from the UK, records a positive response rate to PPD of 16% in paediatric cases [9] and PPD was found to be within the top ten most common allergens in 10 studies of patch testing in children [4]. Temporary black henna tattoos are a common cause of PPD sensitisation in children [20], however these tattoos are not common in Laos and upon questioning, all three children had been directly exposed to PPD from a very young age through hair dye use by one or both parents. Hopefully by explaining the risk of hair dye/PPD exposure to the parents of children with AD attending the allergy clinic, the risk of sensitisation can be reduced.

Other than a diagnosis of AD, no further selection was made regarding the likelihood of allergic contact dermatitis (ACD) in our cohort and the majority (80%) of our participants were under the age of 5 years. As a result, direct comparison with other studies looking at ACD in the paediatric AD population may not be feasible. Despite our cohort not being specifically selected as having possible contact allergy and given that the mean age was only 4.5 years, the data still reveal a fairly high rate of contact sensitivity. The rate of gold and nickel sensitivity in this study are similar to the preliminary results from our sister study looking at contact sensitivity in healthy medical students in Laos. Interestingly, Belloni Fortina et al (2015) [21] considered paediatric patch test results across Europe and found that the highest sensitization rate was in the age group of 1–5 years (45.3% compared to 33.3% between ages 6–12 years). In a further study, Belloni Fortine et al (2011) [22] looked at contact sensitization in children aged between 3 and 36 months and found an even higher rate of positive reactions (62.3%), this was in a cohort of children both with and without atopic dermatitis but all suspected of having contact dermatitis. Both of these papers highlight that contact sensitization is high in the very young, and the results of our study fit with this data.

There are several weaknesses in this study. Firstly, the age of the cohort was very young; this was not intentional but our cohort came from patients attending the allergy clinic and the majority of patients attending this clinic during the study period were younger children. The reasons for this have not been investigated but are possibly due to older children having to miss school to attend clinic, eczema being more prevalent in young children or more parental concern over illness in younger children. Secondly, we used all 3 panels of TRUE Test® series, which ensured consistency in dosage but compared to the British Standard Series, the TRUE Test® series does not include p-Chloro-m-cresol, cetearyl alcohol, sodium metabisulfite, fusidic acid, chloroxylenol, compositae, primin, fragrance mix II, kathon CG, methylisothiazolinone, lyral, limonene or linalool, so any sensitivity to these allergens would have been missed. This study has considered contact sensitivity in the paediatric AD population, it would also have been very interesting to patch test healthy controls. This was not done in this study due to the limited number of patch tests available and it was felt that children with AD may yield more positive results than healthy controls, in addition it is important to consider the risk of sensitization. Finally, no assessment was made regarding the relevance of positive patch tests in this study. Patients and their caregivers were advised to avoid allergens for which they had tested positive but no follow-up assessment of the impact of allergen avoidance for the purposes of this study was made.

## Conclusion

The aim of this study was to document common cutaneous allergens in the paediatric Lao population. Paediatric patients with eczema were used as it was felt that these individuals may have had more exposure to allergens than healthy controls and their attendance at clinic

facilitated their enrolment into the study and attendance at follow-up. The most common allergens resulting in positive patch tests were; gold (18%), nickel (10%), formaldehyde (6%), p-Phenylenediamine (6%), potassium dichromate (2%), cobalt (2%), 2-Bromo-2-nitropropane-1,3-diol (2%), paraben mix (2%), fragrance mix 1 (2%) and neomycin (2%). The majority of the patients with positive reactions were female. It is hoped that the results of this study will help clinicians to advise patients and their parents of common allergens to try and avoid and to inform future work on patch testing in the Lao population.

## Supporting information

**S1 Fig. Patient selection flowchart.**
(DOCX)

**S1 Table. TRUE Test® allergens.**
(XLSX)

**S1 File. Data table for patch testing in paediatric patients with atopic dermatitis in Laos.**
(XLSX)

## Acknowledgments

We would like to express our thanks to the children and their guardians who participated in this study. We are also very grateful to the directors of Children's Hospital for their technical advice.

## Author Contributions

**Conceptualization:** Catriona I. Wootton, Somxay X. Billamay, John S. C. English, Mayxay Mayfong.

**Data curation:** Catriona I. Wootton.

**Formal analysis:** Catriona I. Wootton, John S. C. English.

**Funding acquisition:** Catriona I. Wootton.

**Investigation:** Catriona I. Wootton, Mong K. Sodaly, Somxay X. Billamay.

**Methodology:** Catriona I. Wootton, Mong K. Sodaly, Somxay X. Billamay, John S. C. English, Mayxay Mayfong.

**Project administration:** Catriona I. Wootton, Mong K. Sodaly, Somxay X. Billamay, Mayxay Mayfong.

**Resources:** Catriona I. Wootton.

**Software:** Catriona I. Wootton.

**Supervision:** Catriona I. Wootton, John S. C. English, Mayxay Mayfong.

**Validation:** Catriona I. Wootton.

**Visualization:** Catriona I. Wootton.

**Writing – original draft:** Catriona I. Wootton.

**Writing – review & editing:** Mong K. Sodaly, Somxay X. Billamay, John S. C. English, Mayxay Mayfong.

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
