## [Decision Letter · Decision Letter 0]

3 Jan 2020

PONE-D-19-33939

Patch test results in paediatric patients with atopic dermatitis

PLOS ONE

Dear Dr Wootton,

Thank you for submitting your manuscript to PLOS ONE. After careful consideration, we feel that it has merit but does not fully meet PLOS ONE’s publication criteria as it currently stands. Therefore, we invite you to submit a revised version of the manuscript that addresses the points raised during the review process.

Please proofread the paper carefully and use PlosOne style.

We would appreciate receiving your revised manuscript by Feb 17 2020 11:59PM. To enhance the reproducibility of your results, we recommend that if applicable you deposit your laboratory protocols in protocols.io, where a protocol can be assigned its own identifier (DOI) such that it can be cited independently in the future. For instructions see: http://journals.plos.org/plosone/s/submission-guidelines#loc-laboratory-protocols

We look forward to receiving your revised manuscript.

Kind regards,

Pal Bela Szecsi, M.D. D.M.Sci.

Academic Editor

PLOS ONE

Journal Requirements:

2. Please provide additional details regarding participant consent. Since your study included minors under age 18, please state whether you obtained consent from parents or guardians. Please also specify in your ethics statement whether the ethics committee specifically approved the verbal/oral consent procedure.

Reviewers' comments:

Reviewer's Responses to Questions

**Comments to the Author**

1. Is the manuscript technically sound, and do the data support the conclusions?

Reviewer #1: Yes

Reviewer #2: Partly

2. Has the statistical analysis been performed appropriately and rigorously? 

Reviewer #1: Yes

Reviewer #2: Yes

3. Have the authors made all data underlying the findings in their manuscript fully available?

Reviewer #1: Yes

Reviewer #2: Yes

4. Is the manuscript presented in an intelligible fashion and written in standard English?

Reviewer #1: Yes

Reviewer #2: No

5. Review Comments to the Author

Reviewer #1: The manuscript is in conditions to be publicated. The language is in intelligible English. No concerns about ethics.

The data supports the conclusions. The statistical analysis been performed appropriately.

Reviewer #2: The paper is interesting as it describes results of patch tests in paediatric patients with atopic dermatitis in Vientiane.

However, several aspects of the paper should be revised. For example, the list of allergens should be completed adding the concentration of allergen and the specific substance in which the allergens are diluited.

Furthermore, paper should be revised by a native English speaker as several grammatical errors are present.

6. PLOS authors have the option to publish the peer review history of their article (what does this mean?). If published, this will include your full peer review and any attached files.

Reviewer #1: No

Reviewer #2: No

---

## [Author Response · Author response to Decision Letter 0]

14 Jan 2020

Response to reviewers:

Journal Requirements:

I believe the manuscript is now complying with PLOS One’s style requirements.

2. Please provide additional details regarding participant consent. Since your study included minors under age 18, please state whether you obtained consent from parents or guardians. Please also specify in your ethics statement whether the ethics committee specifically approved the verbal/oral consent procedure.

The text has been adjusted accordingly and now reads:

This study ran from August 2017 to January 2018. Ethical approval was granted by the Lao National Ethics Committee for Health Research. The process of patch testing was explained and images of patch tests being performed were also shown to the patient and their family and verbal consent given by the parent or guardian of the child. Verbal consent was deemed appropriate by the National Ethics Committee for the study as this is standard practice for patch testing in the author’s experience and consent was recorded on the study proforma at each visit. 

This has now been amended, thank you.

Reviewers' comments:

Reviewer's Responses to Questions

5. Review Comments to the Author

Reviewer #1: The manuscript is in conditions to be publicated. The language is in intelligible English. No concerns about ethics.

The data supports the conclusions. The statistical analysis been performed appropriately.

Thank you, we appreciate these comments.

Reviewer #2: The paper is interesting as it describes results of patch tests in paediatric patients with atopic dermatitis in Vientiane.

However, several aspects of the paper should be revised. For example, the list of allergens should be completed adding the concentration of allergen and the specific substance in which the allergens are diluited.

The vehicle for each allergen has now been added, the concentration of each allergen is listed.

Furthermore, paper should be revised by a native English speaker as several grammatical errors are present.

Thank you for your comments, unfortunately despite being a native English speaker, and reviewing the paper with several colleagues, we are able to identify where the grammatical errors mentioned are. Further details on the areas of concern would be greatly appreciated.

---

## [Decision Letter · Decision Letter 1]

25 Mar 2020

Patch test results in paediatric patients with atopic dermatitis in Laos

PONE-D-19-33939R1

Dear Dr. Wootton,

We are pleased to inform you that your manuscript has been judged scientifically suitable for publication and will be formally accepted for publication once it complies with all outstanding technical requirements.

With kind regards,

Pal Bela Szecsi, M.D. D.M.Sci.

Academic Editor

PLOS ONE

Additional Editor Comments (optional):

Reviewers' comments:

Reviewer's Responses to Questions

**Comments to the Author**

1. If the authors have adequately addressed your comments raised in a previous round of review and you feel that this manuscript is now acceptable for publication, you may indicate that here to bypass the “Comments to the Author” section, enter your conflict of interest statement in the “Confidential to Editor” section, and submit your "Accept" recommendation.

Reviewer #2: (No Response)

2. Is the manuscript technically sound, and do the data support the conclusions?

Reviewer #2: Yes

3. Has the statistical analysis been performed appropriately and rigorously? 

Reviewer #2: Yes

4. Have the authors made all data underlying the findings in their manuscript fully available?

Reviewer #2: Yes

5. Is the manuscript presented in an intelligible fashion and written in standard English?

Reviewer #2: Yes

6. Review Comments to the Author

Reviewer #2: (No Response)

7. PLOS authors have the option to publish the peer review history of their article (what does this mean?). If published, this will include your full peer review and any attached files.

Reviewer #2: No

---

## [Editor Report · Acceptance letter]

30 Mar 2020

PONE-D-19-33939R1 

Patch test results in paediatric patients with atopic dermatitis in Laos 

Dear Dr. Wootton:

I am pleased to inform you that your manuscript has been deemed suitable for publication in PLOS ONE. Congratulations! Your manuscript is now with our production department. 

With kind regards,

on behalf of

Dr. Pal Bela Szecsi 

Academic Editor

PLOS ONE